# Saudi secondary school science textbooks' ability to inculcate a PISA-Informed Scientific Identity

Wadha H. Alotaibi[1]\*, Amani Khalaf H. Alghamdi [2]

1 Professor of Curricula and Methods of Teaching Science, Department of Teaching and Learning, College of Education and Human Development, Princess Nourah bint Abdulrahman University, Riyadh, Saudi Arabia, 2 Professor of Curriculum and Pedagogy, Department of Curriculum and Instruction, College of Education, Imam Abdulrahman Bin Faisal University, Dammam, Saudi Arabia

\* whalotaibi@pnu.edu.sa

## Abstract

Saudi Arabia's low performance in the 2018 Programme for International Student Assessment (PISA) science tests ($M = 386$ compared to $M = 489$ overall) prompted a content analysis of Saudi secondary school science textbooks ($N = 9$, three books for each of Grade, 7, 8, and 9) (86% intracoder reliability). Using a definition of *scientific identity* formed using PISA's conceptual framework, we recommend future textbook revisions that ensure more balanced coverage to offset the current uneven weighing of scientific knowledge (55%), scientific competencies (28%), and science context (17%). Right now, students receive substantial coverage of major facts, concepts, and theories (knowledge) but less so for competencies (e.g., proficiency in scientific explanations, design, evaluation, and interpretation) or for how to understand and use science in different contexts at different levels (i.e., personal, local, and global). We assumed that secondary students' appreciation that "science is for me" (i.e., positive scientific identity) will happen, if they are evenly exposed to all PISA dimensions. Because grade level profiles were very similar (both what was present and missing), rebalancing must occur in all three grades. Revised science textbooks that augment current weak spots should positively influence Saudi secondary student's scientific identity and improve PISA performance.

## Introduction

The Organization for Economic Cooperation and Development (OECD) launched the Programme for International Student Assessment (PISA) in 2000. PISA aims to assess the ability of 15-year-old students (Grade 9) to use their knowledge and skills in reading, mathematics, and science to encounter life challenges. The test is administered once every three years based on unified criteria such as students' ages, regardless of their classes, applied questions, and impartiality of history and local,

**Data availability statement:** All relevant data are within the paper and its Supporting information files.

**Funding:** This research project was funded by the Deanship of Scientific Research, Princess Nourah bint Abdulrahman University, through the Program of Research Project Funding After Publication, grant No (44- PRFA-P- 72).

**Competing interests:** The authors have declared that no competing interests exist.

cultural factors. Due to COVID, PISA 2018 did not happen until 2022 [1]. PISA has six performance (proficiency) levels ranging from low 1 to high 6. Higher levels 5 and 6 are more competent in abstract thinking, reasoning, argumentation, complexity, and dealing with uncertainty [1].

With its international framework, PISA is a valuable international study in the education field [2]. Administered by OECD, it focuses on 15-year-old students' (Grade 9) learning, which is a significant indicator of their performance at age 19 when they traditionally enter the work force or enroll in higher education [3]. PISA test results consistently show high and accurate confidence when measuring students' abilities [4,5]. And PISA results are not affected by cultural factors or the surrounding environment of participating countries [6].

We are interested in the science aspect of the PISA because, as we will discuss, Saudi Arabian students' performance was unsatisfactory in 2018. Fig 1 summarizes the three main dimensions of PISA's science test in place for the 2018 test: scientific knowledge, scientific competencies, and scientific context [7,8]. We draw on these to define and operationalize *scientific identity*. We propose that a scientific identity reflects the learner's ability and willingness to use their scientific knowledge and competency in practical situations in different contexts; they can identify with these three aspects of science and integrate them into their persona. For us, science context is an important part of scientific identity because "science is used within a social and environmental context that involves consideration of economics, social behaviour, and [politics]" [1, p. 3].

The OECD [7] identified three contextual levels (i.e., personal, local/national, and global) and five specific contexts: health and disease; natural resources (e.g., conservation and responsible distribution); environmental quality (e.g., sustainability, pollution, and waste management); hazards and risks (e.g., lifestyle, changes to the earth and climate); and the frontiers (leading edge research) of science and technology. The OECD [7] chose these because (a) they are most relevant to Grade 9 students' lives and interests and (b) students' ability to bring their knowledge and competencies to these contexts enhances and sustains a nation's quality of life and relevant policy development. We worked up examples of scientific context and application for all three levels for use later in the study (see Table 1) [1,5,9,10].

### Saudi Arabia's PISA participation

Saudi Arabia's first PISA participation was in 2018, which is relatively late compared to other Arab nations. Jordan, Qatar, and Tunisia started in 2006. Saudi's 2018 PISA mean scores for reading, mathematics, and science were lower than overall performance for all nations ($N = 79$): Saudi reading $M = 399$ compared to $M = 487$; mathematics $M = 373$ compared to $M = 487$; and science $M = 386$ compared to $M = 489$ [11,12]. Most (38%) Saudi students achieved Level 2 or higher, which means they can provide scientific explanations in familiar contexts or draw conclusions based on simple investigations. However, none were proficient at Levels 5 or 6. These students can creatively and autonomously apply their science knowledge in a wide variety of situations, including unfamiliar ones [1].

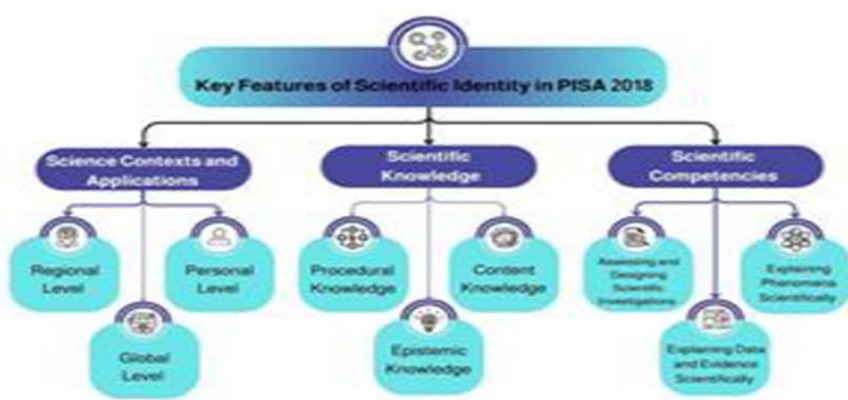

**Fig 1. Key dimensions of scientific identity informed by PISA's 2018 science test.**

**Table 1. Examples of the scientific context and practical application aspect of scientific identity.**

| Scientific Application | Contexts | | |
|---|---|---|---|
| | Personal | Regional (Local/National) | Global |
| **Environmental Quality** | Protecting the environment through sustainably using raw materials and tools and going down the path that leads to environmental sustainment. | Knowing regional environments, available natural resources, and ways of protecting them. | Realising diverse biological variations, global environmental balance, and allies of dealing with global environmental problems. |
| **Health and Disease** | Adopting the required behavior for self-preservation, having good nutrition, and avoiding accidents and diseases. | Protecting the health of society in line with the nature of disease spread and having good health in society. | Dealing with epidemics spread worldwide and taking procedures to reduce their spread. |
| **Risks and Hazards** | Determining and avoiding risks that surround people at the personal level. | Dealing with risks that surround society (e.g., earthquakes, floods, and desertification). | Changes occur in the world because of risks of war, high temperatures, and climate change. |
| **Using Science and Technology** | Mastering technological skills in line with individuals' hobbies and skills. | Using equipment and techniques of a positive effect in society. | Using related scientific equipment in discovering the universe, organisms, and diseases. |
| **Natural Resources** | Personal consumption of natural resources and energy. | Protecting natural resources in society, population structure, and appropriate resource distribution. | Achieving sustainable development and protecting global resources to achieve a global balance between renewable and nonrenewable resources |

## Literature review

Improving Saudi Arabia's science PISA performance will require work on all curricular products. In addition to curriculum, Depren [13] and Gunaydin and Basaran [14] found that students' reading comprehension, scientific content knowledge, attitude toward science, and the type of assessment and evaluation tool influenced their PISA performance.

Regarding the science curriculum itself, Cansiz and Cansiz [15] attributed Turkish students' low PISA science performance to the nature of the science curricula, which insufficiently reflected the scientific identity dimension of PISA tests (see Fig 1). Al-Towaisi [16] confirmed that curricular content affected students' PISA test performance followed, respectively, by educational policies, teacher performance, student self-competence, and schools' surrounding environment. Al-Bashabesha [17] and Mubarak [18] similarly argued that students' PISA science test performance was dependent on the nature and content of science curricula. Hepworth et al. [19] argued that Irish students performed better on PISA than students from the other three United Kingdom countries because of Ireland's science curricula excellence.

Studies have shown that well-developed instructional materials (including textbooks) can also improve students' PISA performance in science and mathematics above and beyond the caliber of the entire curriculum [10]. Given Egyptian students' low PISA performance, Abdulfattah [20] recommended developing science textbooks that consider PISA frameworks. Turk et al. [21] blamed Turkey's low 2015 PISA performance on the textbooks. Students performed at PISA levels 1–4 but not the two higher levels. Turkish science textbook content focused on Bloom's three lower levels of learning (remember, understand, and apply – not analyze, evaluate, and create). Sothayapetch et al. [22] found that Thailand focused on the scientific method (process) while Finland focused on scientific concepts and contexts. Finland always scores higher than Thailand on PISA. They recommended focusing on and analyzing science textbooks to discern scientific facts, concepts, context, theories, and contexts.

Al-Zahrani [12] offered suggestions for improving Saudi Arabia's unsatisfactory PISA performance: redevelop curricula, augment teachers' performance, improve students' willingness to learn science, and prepare learning environments conducive to science learning. But given the research affirming the powerful role that textbook content plays on PISA performance, we were interested in how well existing Saudi secondary science textbooks can contribute to students gaining a scientific identity, so they can prepare for the future and perform better on PISA.

## Scientific identity

Saudi's less-than-satisfactory performance on the last PISA test, especially in the science realm, prompted our concern. It seems Saudi students were not identifying with science in their lives. We are convinced that gaining information about the content of Saudi textbooks relative to PISA's test parameters (see Fig 1) would serve as a primary reference point for a series of procedural steps to prepare Saudi students for future PISA tests and as Saudi and global citizens.

We were encouraged by several studies that reported that students' acquisition of a Science, Technology, Engineering, and Mathematics (STEM) identity had a positive effect on their desire to continue studying STEM topics, transferring learning into their professional life, and mastering scientific skills and research motivation [23–26]. We reasoned that a person with a *scientific identity* is also more inclined to participate in activities of a scientific community and appreciate the practical and functional value of science (per [25]).

Generally, a person's scientific identity affects their scientific activity, the development of professional goals containing science, and their future perceptions of science. Individuals acquire a scientific identity through iterative interactions with scientific contexts and social and functional situations. Once attained, they show positive attitudes toward scientific practices in a way that reinforces a sense of belonging, pride, and high estimations of scientific knowledge and skills [27–29]. As a caveat, we are aware that the OECD [8] recently expounded on how it conceptualizes a scientific identity, but this approach was not employed in our study, which drew instead on the PISA framework in place and intended for the 2018 testing event [7].

We concur with the OECD's [8] position that "scientific identity is fundamental to science learning and attainment" (p. 12). We just defined and operationalized it differently in our study. We said scientific identity (i.e., whether people identify with science) depends on students' scientific knowledge, competencies, and context per the OECD's 2019 framework [7]. In contrast, the OECD [8] said scientific identity is separate from scientific knowledge, competencies, and context. Scientific identity encompasses, instead, (a) the attitude that "science is for me" (science capital). (b) Students would have critical science agency – they would use science to address social inequalities. (c) They would view science as an ethics- and value-ladened practice. (d) And they would appreciate that science learning depends on inclusive and diverse representations and experiences of science [8].

## Problem statement, research question, and study objectives

Saudi Arabia's unsatisfactory performance on PISA tests in general, and science in particular, demands strategic efforts for improvement especially science curricula content and instructional materials like textbooks. Research shows that students' performance on PISA tests depends on the nature of national science curricula that they experienced before

completing the test. Low performance on the PISA science test is indicative of an ill-developed scientific identity. This suggests that inadequate textbook content regarding scientific knowledge, competencies, and real-world contexts hinders the development of a scientific identity in Saudi Grade 9 students. If so, there is a need to determine the extent to which existing textbooks contain content pursuant to a scientific identity as defined in this study. To the best of our professional and scholarly knowledge, research in this area is lacking in Saudi Arabia.

Also, although PISA now measures scientific identity as of 2020, students who strongly identify with science because of their secondary schooling and textbook content should perform better on PISA because they go into the test already valuing science [30]. The main research question was thus, "*To what extent do Saudi secondary school science textbooks include s*cientific *identity to prepare students for the future?*" Study objectives (i.e., tasks to collect data to answer the research question) included

1. To determine the key dimensions of the PISA science test;

2. To conduct a content analysis of Saudi secondary level science textbooks issued in 2022–2023 to determine the presence of key PISA science dimensions; and

3. To judge the ability of Saudi secondary science textbooks to help students gain a scientific identity.

**Study significance.** Using our empirical evidence of the extent to which building a scientific identity is present in Saudi secondary science textbooks, we intend to (a) propose a strategic vision around bolstering current levels of inclusion, (b) contribute to the strategic trend of developing Saudi science textbooks that entrench aspects of PISA's framework, (c) support Saudi education officials in efforts to improve the Kingdom's PISA performance and (d) affirm the scientific value of likeminded research for reading and mathematics. Developing Saudi curricular content so that it aligns with PISA test content applies to science, reading, and mathematics [11,31–33].

Conducting research about a nation's PISA performance is beneficial on several fronts. Research can generate information useful for curricular revisions and preparing students for future PISA tests. Said research can inform government-level development plans as well as changes to the education system. Research can inform efforts to create a school culture that supports students' PISA participation and, on a larger scale, socializes students to be active citizens who use science to paint the state in a positive light. And attendant research results can help students gain a scientific identity they can take into the work force [1,13,14,16].

## Method

The descriptive-analytical approach we employed is well-suited for analyzing the content of science textbooks intended for Saudi Arabia's secondary school level. We strove to determine the extent to which the textbooks' content helped students gain a scientific identity as we defined it using PISA's [7] three science dimensions: scientific knowledge, competencies, and context.

### Document sample frame

The sample frame comprised $N=9$ approved science textbooks in use at the time of the study (three for each of Grade 7, 8, and 9). Although only Grade 9 students take the PISA test [1], we sampled all three levels as a preemptive strategy because, in addition to knowing what Grade 9 students were learning just before taking the PISA test, insights gained about the lower grades may prove useful in any future changes to Saudi secondary school science textbooks' content.

### Study procedures

To address the research question effectively, we adhered to the following structured steps: (a) identified PISA test dimension indicators, (b) developed and validated a content analysis tool and (c) applied the tool to conduct a content analysis of the document sample frame.

*Identify PISA scientific identity dimensions.* As noted in the literature review (see Fig 1 and Table 1) we used PISA's [7, p. 103] three scientific dimensions to define scientific identity: (a) *scientific knowledge* (conceptual, procedural, and cognitive/epistemic); (b) *scientific competence* (explain scientific phenomena, evaluate and design scientific inquiry, and interpret scientific data and evidence); and (c) the *context of science* and its practical applications (including health and disease, natural resource utilization, environmental considerations, potential risks, and the frontiers of science and technology).

**Develop and validate content analysis tool.** Using the information gained from step 1, we developed a content analysis coding tool that incorporated a comprehensive list of indicators associated with the PISA 2018 and PISA 2024 test dimensions [7, chapter 4] [8]. To validate the tool and ensure its accuracy, a group of science curriculum and teaching methods experts assessed the correlation of the indicators to the field it contained, integrity of linguistic formulation, and any modifications that would serve the study from their point of view. We heeded their observations, which included merging some indicators, separating others, deleting any duplicate or overlapping indicators, and modifying some language formulations. A satisfactory intracoder reliability coefficient (86%) was achieved with one researcher using the tool to analyze a science textbook sample twice within an interval of four weeks [34].

## Data analysis

Content analysis entails counting things and placing them in categories [34]. The units of analysis within the documents included phrases, sentences, and paragraphs (including a series of paragraphs). As the content of each document was read, and data were coded using the coding sheet (i.e., validated content analysis tool), data points were assigned to one of the three PISA dimensions: scientific knowledge, competencies, and context. Column and row frequencies and percentages were calculated for each document. Results are reported using descriptive statistics (frequencies and percentages).

## Results and discussion with recommendations

The research question was "*To what extent do Saudi secondary school science textbooks include scientific identity that prepares students for the future?*" In this study, we operationalized scientific identity as comprising PISA's scientific knowledge, competencies, and context [7]. Table 2 profiles the overall content analysis results. The science textbooks contained mainly scientific knowledge (55%) followed by scientific competencies (28%) and science context and practical applications (17%). Discussion points and recommendations are integrated into presentation of results.

To answer the research question, the heavy incidence of one PISA dimension relative to the other two suggests that current textbooks lack the ability to ensure that Saudi secondary students gain a full scientific identity. They would be substantially exposed to major facts, concepts, and theories but less so for competencies (proficiency at scientific explanations, design, evaluation, and interpretation) and for how to understand and use science in different contexts at different levels. This imbalance might contribute to Saudi Grade 9 students saying, "Science is *not* for me." If this is the case, it could explain their unsatisfactory PISA science performance in the 2018 test rounds ($M = 386$ compared to the $M = 489$ international average) [11,12].

**Table 2. Evidence of PISA's three dimensions in saudi secondary school science textbooks.**

| PISA Aspect of Scientific Identity: | Grade 7 | | Grade 8 | | Grade 9 | | Totals | |
|---|---|---|---|---|---|---|---|---|
| | *n* | % | *n* | % | *n* | % | *N* | *%* |
| **Scientific Knowledge** | 484 | 30 | 551 | 34 | 578 | 36 | 1613 | **55%** |
| **Scientific Competencies** | 276 | 33 | 266 | 32 | 288 | 35 | 830 | **28%** |
| **Scientific Context** | 122 | 25 | 211 | 43 | 154 | 32 | 487 | **17%** |
| **Total** | 882 | **30%** | 1028 | **35%** | 1020 | **35%** | 2930 | 100% |

## Scientific knowledge

Scientific knowledge includes content, procedural, and epistemic knowledge. Content knowledge pertains to concepts, facts, theories, models, and ideas about the natural world that science has already established. Procedural knowledge concerns procedures that scientists use to apply the scientific method (empirical inquiry) to discover and establish the content knowledge. Epistemic or cognitive knowledge refers to what is involved in, and what are the functions of, questioning, observing, theorizing, hypothesizing, modelling, and argumentation. It also involves appreciating the different forms of scientific inquiry and the role impartial peer review plays in determining if content knowledge is true and can be trusted [7].

Per Fig 2, nearly two thirds (58%) of the scientific knowledge dimension in Saudi secondary science textbooks pertained to content knowledge followed by procedural knowledge (25%) and cognitive/epistemic knowledge (17%). This result affirms that all three forms of scientific *knowledge* were present in the textbooks, but representation was uneven. Al Hawr [35] found a similar pattern for Palestinian textbooks.

Content knowledge in the science textbooks focused mostly on (a) *physical and chemical systems* (42%) (especially motion and force, and physical properties [50%]); (b) *living systems* (40%) (especially cell chemistry, genetics and biodiversity, conservation, and species classification [51%]); and (c) *earth and space systems* (18%) (especially plate movement, and earth's structure [79%]). Procedural knowledge mostly covered scientific experiments (42%) and data collection and representation (32%). Epistemic knowledge overwhelmingly (80%) dealt with drawing deductive and inductive scientific conclusions.

These results align with Al-Fahidi's study, which identified physical and chemical systems as the most recurrent areas in Saudi secondary science books followed by living systems, and then Earth and space systems. Similarly, in a Palestinian study, Al Hawr [35] ranked physical and chemical systems at the top with variations in the rankings of living systems, and earth and space systems. Our results suggest, however, that when taught using the existing textbooks, Saudi secondary science students would miss out on certain aspects of scientific knowledge (low percentages 4–10%): research variables; main components of science (concepts, laws, theories, and models); formulating assumptions; acids and bases; energy interactions; light and sound; structure of organisms; food chains; history of the earth; and history of the universe. These components should be strengthened in future textbook revisions.

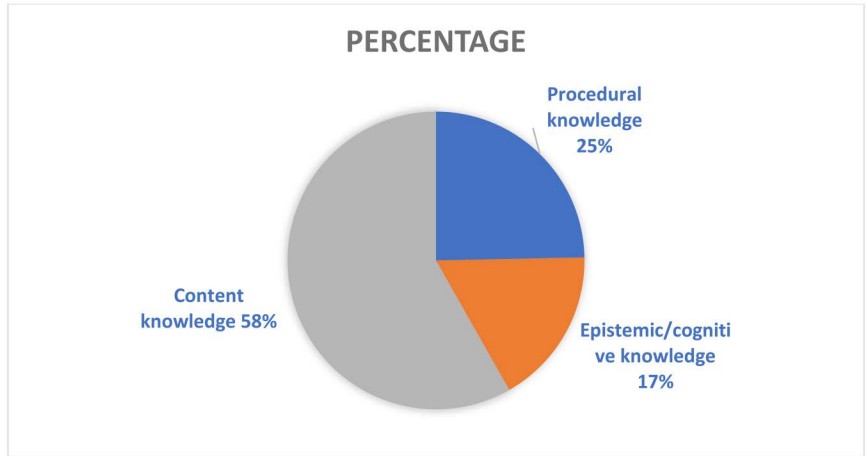

**Fig 2. Percentage of PISA's three scientific knowledge components in Saudi secondary science textbooks.**

## Scientific competencies

The OECD [7] clarified that its measurement of *scientific competency* reflected the scientific knowledge (i.e., content, procedural, and epistemic) "that 15-year-old students can reasonably be expected to have" (p. 101). Scientific competency comprises three factors: "the ability to explain phenomena scientifically, evaluate and design scientific enquiry, and interpret data and evidence scientifically" (p. 102). Respectively, students must be able to access and apply the scientific knowledge they hold to (a) recognize, offer, and evaluate scientific explanations for a range of natural and technological phenomena; (b) describe and appraise scientific investigations and inquiries (e.g., measurements, reliability, validity, and statistical confidence) and propose alternative ways to scientifically address questions; and (c) critically (using a skeptical disposition) analyze (look for patterns), evaluate (judge), and interpret (draw conclusions) scientific data, arguments, and claims from a variety of sources that employ a range of representations.

Scientific competencies in the Saudi secondary science textbooks focused equally on the ability to explain phenomena scientifically (46%) and interpret data and evidence scientifically (43%) and less so on evaluate and design scientific enquiry (11%) (see Fig 3). This result affirms that all three forms of scientific competencies were present in the Saudi textbooks, but representation was skewed to two components (89%). Al Hawr [35] reported similar results in Palestinian textbooks, wherein the most recurrent scientific competencies were the ability to elucidate scientific phenomena followed very closely by the aptitude to clarify data and scientific evidence and to a lesser extent the capability to assess and design scientific inquiry. Future Saudi textbook revisions should bolster the *evaluate and design scientific inquiry* component of scientific competencies by highlighting a critical and skeptical disposition and helping students to analyze to find patterns, make judgements, and draw scientific conclusions [7].

To continue, the *scientific interpretation of phenomena* competency component focused mostly on providing an explanation of scientific phenomena (35%) followed by equal coverage (average 21% each) of linking causes and consequences, applying knowledge in real life situations, and predicting changes. The *scientific interpretation of data and evidence* component provided fairly equal coverage of proposing alternative ways to address questions scientifically (34%), articulating tradeoffs between alternatives (31%), and generalizing answers to deal with specific questions (28%). The *assessment and design of scientific investigations* component focused equally on drawing conclusions (45%) and justifying data interpretation (43%).

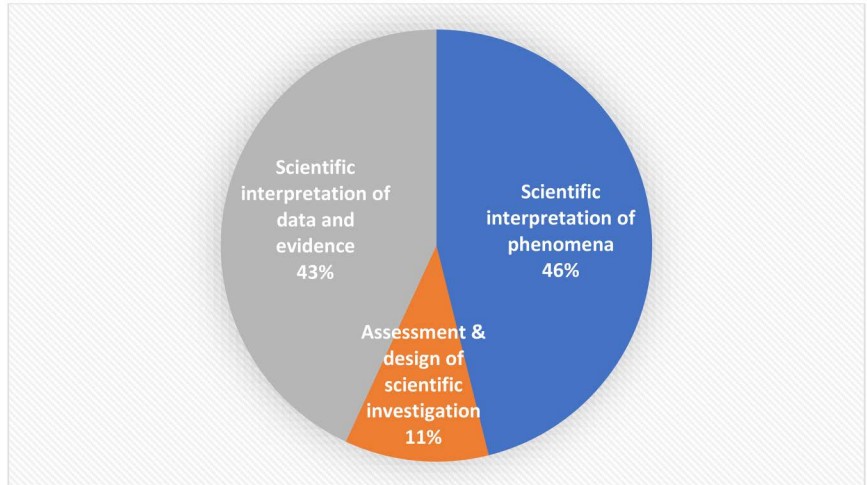

**Fig 3. Percentage of PISA's three scientific competency components in Saudi secondary science textbooks.**

We believe the organizational format of the Saudi textbooks contributed to the high presence on some PISA elements [7]. Each book contained sections titled *unit projects, science notebook, the experiment, initiated experience, what did you read,* and *investigation from real life.* If Saudi secondary science students complete these learning tools, they, respectively, gain experience expounding on topics found on the internet, recording findings about phenomena in their science notebook, applying analytical skills to derive conclusions from their experiments, employing critical thinking to dissect and elucidate their results, deducing information and proffering scientific explanations for phenomena they have read about, and conducting real-life inquiries. Saudi secondary science teachers should continue to employ these practices on a regular basis and expand their usage to all aspects of the curriculum. Their use seemed to strengthen the possibility of developing a scientific identity.

That said, when taught using the existing textbooks, Saudi secondary science students would miss out on certain aspects of scientific competencies in each of the three components (low percentages 1–10%): distinguish between scientific arguments (1%), formulate hypotheses (2%), ask precise scientific questions (7%) and assess the validity of results (11%). These components should be strengthened in future textbook revisions. They require Bloom's higher levels of cognitive learning (i.e., analysis, evaluation, and creation), which Turk et al. [21] also found lacking in secondary science textbooks. This result may explain why Saudi students did not score at levels 5 and 6 in the 2018 PISA test [8].

## Scientific context

Our conceptualization of scientific identity included *scientific context* and its practical applications. The OECD [7] viewed scientific context as current and historical issues that demand some understanding of science and technology (see five contextual issues at Table 1). Also, "items in the PISA 2018 science assessment may relate to the self, family and peer groups (personal), to the community (local and national) or to life across the world (global)" [7, p. 103]. Instead of assessing the context itself, the PISA test assesses students' scientific knowledge and scientific competencies in specific contexts.

This means Saudi secondary science textbooks must include reference to these three levels and five contexts if students hope to pass this part of the PISA test and gain a richer sense of scientific identity as defined in our study. The content analysis revealed that although all three levels and all five contexts were evident in the textbooks, coverage focused mainly on two contexts: frontiers of science and technology (37%) and health and disease (30%) totaling more than two thirds (67%) (see Fig 4). The quality of the environment (15%), natural resources (11%) and risks and hazards (7%) received the least coverage.

We attribute the highest percentage for frontiers of science and technology (37%) to the textbooks' significant emphasis on developing students' research skills through the utilization of search engines and their ability to draw conclusions and acquire information across various fields. This emphasis is particularly evident in *Unit Projects*, which revolve around online research, and *Science via Websites*, which conclude each chapter and play a crucial role in reinforcing these skills. The emphasis on health and disease (30%) likely reflects the textbooks' constant reference to students adhering to safety guidelines: maintaining cleanliness during experiments, safeguarding hands and eyes, and handling tools and materials with utmost care. These are reiterated in each chapter via sections on *initial experiment, experiment,* and *investigation from real life.* Indeed, most of the health and disease context scored for health (82%) and not disease (18%).

***Three contextual levels.*** Regarding the three contextual levels, the majority (59%) of textbook content focused on the personal level (self, family, and peer groups), followed by global (24%) and local/national (community) (17%). This result suggests that the *scientific identity* of Saudi secondary science students is skewed to the personal level. They may not enter high school with the sense that they need scientific knowledge and competencies to deal with local/national (regional), and global issues with scientific overtones. They may not identify with the local, national, regional, and global levels from a scientific perspective. Taught using existing textbooks, Saudi science students cannot attain a scientific concern for the five contextual areas *beyond the personal*. Future revisions to Saudi secondary science textbooks must redress this issue so students' scientific identity extends to all three levels – personal, local/national, and global.

 

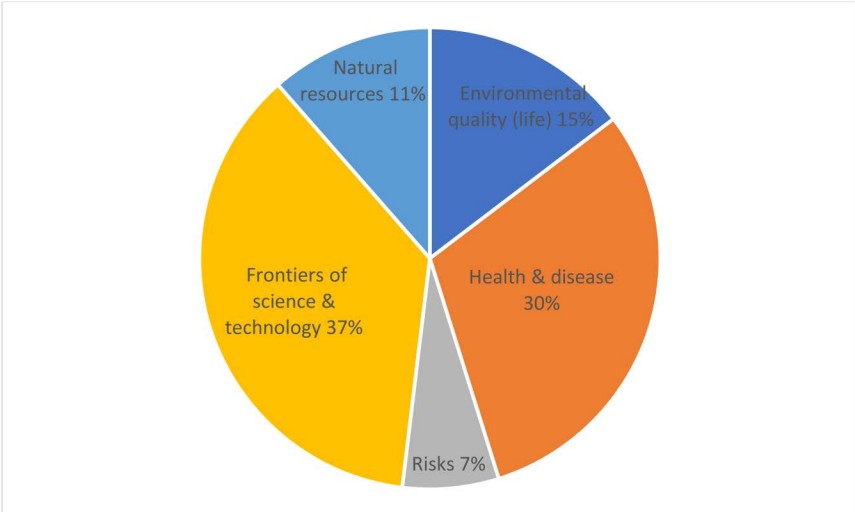

**Fig 4. Percentage of PISA scientific contexts in Saudi secondary science textbooks.**

## Grade level profiles

We collected data for all three secondary science grades even though only Grade 9 students take the PISA test. Results showed that the profile for each Saudi grade was very similar (what was present and missing) (see Table 2). Also, evidence of the textbooks' ability to inculcate a scientific identity, as conceptualized for our study, was present in all three grades at virtually the same level with Grade 7 nominally lower on account of less coverage of the scientific context (see Table 2). This result clearly suggests that any revisions to rebalance (accommodate what was missing) must occur in all three grades if inculcating a scientific identity is to succeed.

To that end, results suggest bolstering specific aspects of each separate PISA dimension. The scientific knowledge component could be augmented by adding information about both procedural and epistemic knowledge. Scientific competencies could be enriched by adding more on evaluating and design scientific enquiry. The scientific context dimension could benefit from more information on local and global levels; diseases; natural resources (e.g., conservation and responsible distribution); environmental quality (e.g., sustainability, pollution, and waste management); and hazards and risks (e.g., lifestyle, and changes to the earth and climate). It is imperative that Saudi curricular designers and textbook authors reconsider PISA dimensions with limited recurrence and enhance their presence within future science textbooks.

## Conclusion

The heavy incidence of one PISA dimension relative to the other two suggests that existing textbooks lack the ability to ensure that Saudi secondary students can gain a full scientific identity. We recommend that future Saudi Grade 7, 8, and 9 science textbooks contain more balanced coverage of the three aspects of scientific identity to offset the current weighing of scientific knowledge (55%), scientific competencies (28%), and science context (17%). We reason that an appreciation of "science is for me" (i.e., a positive scientific identity) [8] will happen if learners are evenly exposed to all PISA dimensions. Others concur that improving science textbooks, so they align with PISA, is a recommended strategy for improving test performance [20–22] and, by association, scientific identity. With this new identity, students should be more inclined to partake in Saudi's scientific community and appreciate the value of science in society when they graduate [27,29].

We further maintain that Saudi Arabia's PISA performance should improve over time because revised textbooks would be more aligned with test parameters [12]. Saudi scores for 2018 were in the 300 range [11,12] with 500 the normal PISA

distribution with a standard deviation of 100 points [7]. Also, the weakest textbook parts right now (especially epistemic knowledge) tend to require Bloom's higher levels of learning, which are PISA's higher proficiency levels (5 and 6). Saudi students did not achieve this proficiency on any science dimension [11,12]. Revised science textbooks that augment current weak spots should positively influence Saudi secondary student's scientific identity and improve their PISA performance. This bolstered identity would bode well for their future as well as the nation's.

## Supporting information

**S1 File. Data analysis 2025 July.**
(DOCX)

**S1 Fig. PISA figure.**
(PPTX)

## Author contributions

**Conceptualization:** Wadha H. Alotaibi, Amani Khalaf H. Alghamdi.

**Data curation:** Wadha H. Alotaibi.

**Formal analysis:** Wadha H. Alotaibi, Amani Khalaf H. Alghamdi.

**Methodology:** Wadha H. Alotaibi, Amani Khalaf H. Alghamdi.

**Validation:** Wadha H. Alotaibi, Amani Khalaf H. Alghamdi.

**Writing – original draft:** Amani Khalaf H. Alghamdi.

**Writing – review & editing:** Amani Khalaf H. Alghamdi.

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
