## [Decision Letter · Decision Letter 0]

29 Mar 2025

Dear Dr. Alghamdi,

Thank you for submitting your manuscript to PLOS ONE. After careful consideration, we feel that it has merit but does not fully meet PLOS ONE’s publication criteria as it currently stands. Therefore, we invite you to submit a revised version of the manuscript that addresses the points raised during the review process.

We look forward to receiving your revised manuscript.

Kind regards,

Dawit Dibekulu, PhD

Academic Editor

PLOS ONE

Journal Requirements:

Princess Nourah University

Deanship of Scientific Research

Additional Editor Comments :

The manuscript provides insightful analysis and is suitable for publication with minor revisions. These include enhancing the abstract, improving section transitions, and addressing minor language inconsistencies. The study is a timely and relevant addition to the discourse on science education and curriculum alignment with international benchmarks.

Reviewers' comments:

Reviewer's Responses to Questions

**Comments to the Author**

1. Is the manuscript technically sound, and do the data support the conclusions?

Reviewer #1: Yes

Reviewer #2: Yes

2. Has the statistical analysis been performed appropriately and rigorously?

Reviewer #1: Yes

Reviewer #2: Yes

3. Have the authors made all data underlying the findings in their manuscript fully available?

Reviewer #1: Yes

Reviewer #2: Yes

4. Is the manuscript presented in an intelligible fashion and written in standard English?

Reviewer #1: Yes

Reviewer #2: Yes

Reviewer #1: This submission clearly presents the problem of SA's low PISA scores in science. it mentions the content analysis of textbooks, the sample size the grades and the intra-coder reliability all of which are important. it also has a sound theoretical framework,i.e., PISA's conceptual framework. the findings show the imbalance in textbook coverage and recommends revision. i suggest the author/s to proofread the submission for minor issues which i have mentioned in the attached text. (i wonder if there is the option of uploading the file below).

Reviewer #2: Dear Author,

I would like you to inform the points that would be included in your research:

In the abstract

The recommendation should be clearly stated

specify what revisions might entail

Introduction

Give detail information about the experiment

The conclusion need to be clearly specified and supported with evidences

Problem Statement, Research Question, and Study Objectives

Some concepts need to be explained. For instance, the relationship of scientific identity on the performance of PISA and the strategies that could help to enhance the achievement of PISA are not also explained. Furthermore, the research gaps would be stated clearly.

Method

descriptive analytical and grade 9 selecting reasons should be justified well.

Scientific dimensions of PISA would be explicitly stated to the importance of the issue of scientific identity.

in the recommendation, you need to show scientific identity can engage students in the contemporary era of technology in the future.

**Do you want your identity to be public for this peer review?** For information about this choice, including consent withdrawal, please see our Privacy Policy

Reviewer #1: No

Reviewer #2: No

---

## [Author Response · Author response to Decision Letter 1]

9 May 2025

Attached a detailed letter of responses the first and 2nd

---

## [Editor Report · Decision Letter 1]

14 May 2025

Saudi Secondary School Science Textbooks’ Ability to Inculcate a PISA-Informed Scientific Identity

PONE-D-25-08148R1

Dear Dr. Alghamdi,

We’re pleased to inform you that your manuscript has been judged scientifically suitable for publication and will be formally accepted for publication once it meets all outstanding technical requirements.

Kind regards,

Dawit Dibekulu, PhD

Academic Editor

PLOS ONE
---

## [Editor Report · Acceptance letter]

PONE-D-25-08148R1

PLOS ONE

Dear Dr. Alghamdi,

I'm pleased to inform you that your manuscript has been deemed suitable for publication in PLOS ONE. Congratulations! Your manuscript is now being handed over to our production team.

Kind regards,

on behalf of

Dr. Dawit Dibekulu

Academic Editor

PLOS ONE